# Human Resources for Oral Health Care in South Africa: A 2018 Update

**DOI:** 10.3390/ijerph16101668

**Published:** 2019-05-14

**Authors:** Ahmed Bhayat, Usuf Chikte

**Affiliations:** 1Department of Community Dentistry, School of Dentistry, University of Pretoria, PO Box 1266, Pretoria 0002, South Africa; 2Department of Global Health, Faculty of Medicine and Health Sciences, University of Stellenbosch, PO Box 241, Cape Town 8000, South Africa; umec@sun.ac.za

**Keywords:** workforce, dental

## Abstract

To describe the current oral health care needs and the number and category of dental personnel required to provide necessary services in South Africa (SA). This is a review of the current disease burden based on local epidemiological studies and the number of oral health personnel registered with the Health Professions Council of South Africa (HPCSA). In SA, oral health services are rendered by oral hygienists, dental therapists, dentists, and dental specialists. Dental caries remains one of the most prevalent conditions, and much of them are untreated. The majority of oral care providers are employed in the private sector even though the majority of the population access the public sector which only offers a basic package of oral care. The high prevalence of caries could be prevented and treated by the public sector. The infrastructure at primary health care facilities needs to be improved so that dentists performing community service can be more effectively utilized. At present, SA requires more dental therapists and oral hygienists to be trained at the academic training institutions.

## 1. Introduction

Human resource needs, planning, and management are essential in the delivery of quality health care services. South Africa (SA), like many developing countries, has a severe shortage of health personnel coupled with a high oral health disease burden [1]. Although there are many preventative programs in place, the prevalence of oral diseases amongst young children still remains high and the disease remains largely untreated [2,3]. As a result, one key aspect that needs to be discussed and aligned with the current disease burden is human resources.

In SA there are two broad categories of oral health personnel, namely clinical and non-clinical personnel. The clinical personnel include those who interact directly with the patient and are the oral hygienists, dental therapists, dentists, and dental specialists.

The non-clinical personnel are the dental assistants, dental technicians and administration staff including receptionists and cleaners.

This paper focusses only on the clinical personnel that provide oral health services in SA.

Oral hygienists (OH) can study for three years to earn a Bachelor of Oral Hygiene (OH) degree. The course is currently offered at four dental schools, namely Universities of Pretoria (UP), Sefako Makgatho (SMU), Western Cape (UWC) and Witwatersrand (Wits). The University of Kwa-Zulu Natal (UKZN) intends to train oral hygienists from 2019. The core function of the OH is the prevention of oral disease and the promotion of good oral hygiene practices through diet and tobacco cessation advice [4]. Oral hygienists can interview patients, do oral examinations and make a clinical diagnosis. Oral hygienists are able to take and interpret analog and digital radiographs. During oral examinations, OHs are trained to scale, polish and apply fluoride, pit and fissure sealants, preventative resin restorations, and whitening agents. Oral hygienists are capable of administering topical and local anesthesia, including infiltration and blocks. Oral hygienists can also place and remove orthodontic bands, retainers, and pre-activated orthodontic appliances. Oral hygienists are employed both in the private and public sector and have been allowed to practice independently in the private sector since 2013.

Dental therapists (DT) complete a three-year full-time degree, Bachelor of Dental Therapy (BDT) offered at UKZN and SMU. Their scope of practice includes the duties of the OHs and additional curative services, such as direct restorative procedures, non-surgical removal of teeth, and roots under local anesthesia and treatment of post-extraction complications [5].

Dentists complete a five-year full-time dental degree, Bachelor of Dental Surgery (BDS) or Baccalaureus Chirurgiae Dental (BChD) with an additional year of compulsory community service in an accredited dental setting. Four dental schools offer this degree, namely UP, SMU, UWC, and Wits. Dentists are qualified to do all the procedures that can be done by a DT and OH in addition to fitting removable dentures, fitting fixed prosthesis including crowns and bridges, removing orthodontic appliances, endodontic procedures, prescribing medication for dental related disorders and taking biopsies [6].

Currently, there are six types of dental specialists (DS) in SA: maxillo-facial, prosthodontists, periodontists, oral pathologists, orthodontists and community dentists. Dental specialists have to complete a full-time five- (in pathology) or a minimum of four- (in all other specialty divisions) year Masters of Dentistry Degree (M Dent/MChD) at one of the four dental schools in SA. Currently, in SA, there are 481 DS registered with the Health Professionals Council of South Africa (HPCSA) with different areas of specialization (Table 1) [1].

The most common oral health personnel are dentists (77%) followed by oral hygienists (15%), dental therapists (8%) and dental specialists (5%) [7]. Since the dentists’ primary function is curative care, we could assume that most dental services being rendered by dentists are curative rather than preventive in nature. Although there is a huge demand for curative care in SA, there is an equally important need for preventive care. If most oral personnel are trained to provide curative and rehabilitative care, preventive services may be reduced, leading to an increase in the prevalence of dental caries and other dental diseases. This, together with a lack of equipment at many dental facilities could be the reasons for the high number of teeth being extracted in the public sector facilities [8].

The health workforce in an ‘ideal’ setting should have a triangular distribution of health workers (Figure 1) [9]. The number of providers should be the largest at the base and the fewest at the apex and the type of services being more complex with progression towards the apex. The greatest number of personnel should be OHs who offer preventive services. The next most common cadre should be the dental therapists who can deliver limited curative care at the grass-roots level where the demand is the greatest. Ideally, dental therapists should be encouraged to work in the public sector to ensure that most of the population has access to basic curative dental services. Finally, at the top of the pyramid (Figure 1) are the dentists and dental specialists who focus on dental rehabilitation, including fixed prostheses, orthodontic services, implant placement, and specialized curative care including endodontics, pulpotomies and pulpectomies.

If the bulk of dental diseases, including dental caries, are prevented and treated at the primary health care level, the demand for specialized rehabilitative services should decrease. If this model were followed, the demand for oral hygienists and dental therapists should be much higher than for dentists and dental specialists.

### Delivery of Health Care in SA

The South African health care system is divided into two sectors, namely the public and private sectors. Although the private sector treats approximately only 16% of the population, the majority of health care providers (62% of medical doctors, 66% of specialists and 90% of dentists) are employed in the private sector [10,11,12].

The public sector is a three-tiered system based on the Primary Health Care (PHC) approach. The levels of service include primary, secondary, and tertiary services. The primary level provides preventive services and offers a basic package of oral health care that includes: oral examinations, radiographs, dental extractions and restorations, and antibiotic prescriptions. These services are provided at no direct cost to the patient at most PHC facilities across the country. Most services can be delivered by dental therapists and oral hygienists, except for prescribing antibiotics. If patients require medication, they can be referred to the medical professionals within these facilities.

The secondary level of care is provided at district hospitals. There are fewer district hospitals compared to PHC facilities, and patients are only referred to district hospitals if the PHC facility does not have suitably qualified staff or adequate facilities to treat the condition. District hospitals are staffed by more specialized personnel, usually dentists and dental therapists who are qualified to provide curative care of prevalent diseases. In addition to primary level services, dentists in district hospitals provide removable dentures, removable orthodontic appliances, and extractions of impacted teeth.

The tertiary level of service delivery is provided at academic hospitals and offers highly specialized services, predominantly rehabilitative in nature. The services include orthodontics, prosthodontics, and endodontics which are rendered by dentists, dental specialists or registrars.

In this paper, we review a number of dental professionals registered with the Health Professional Council of South Africa (HPCSA). We consider the different cadres of dental health professionals and compare them to the reported burden of disease in order to identify treatment gaps and make recommendations for provincial health authorities and dental training institutions. The data was obtained from the Health Professions Council of South Africa and academics at the various dental institutions.

## 2. Disease Burden

### 2.1. Dental Caries

Dental caries is the most prevalent oral disease in SA and almost 60% of six-year-olds have had dental caries [13]. Most of the caries are untreated (45–60%) and although school fissure sealant programs are in place, it seems that six-year-olds continue to have a high prevalence of caries [13]. Although the number of health personnel has increased, the disease burden has continued to rise over the past few years [2,3,14]. This could be due to urbanization, which has been shown to be associated with an increase in sugar consumption [15], an increase in the population of young children in the country and a lack of clinical services in some areas. Much of this disease burden could be addressed by appropriate oral hygiene instructions, fluoride applications, and fissure sealants, all of which could be delivered by dental therapists and oral hygienists.

### 2.2. Malocclusions

Few studies have determined the prevalence of malocclusions requiring orthodontic treatment. The National Children’s Oral Health Survey (NCOHS) reported that 32% of 12-year-olds required some form of orthodontic treatment [13]. Most of these cases were associated with the early extraction of primary and permanent teeth probably indicated due to the presence of dental caries [16]. Possibly, the prevalence of malocclusions would have been lower if dental caries were prevented or treated conservatively through restoration rather than extractions.

### 2.3. Periodontal Disease

Similar to the orthodontic needs in South Africa, there is little data on the prevalence and subsequent treatment of periodontal disease. The NCOHS reported that only 15% of 15-year-olds had a healthy periodontium while 60% were diagnosed with calculus [13]. According to the 1988/89 National Oral Health Survey, 67% of adults were diagnosed with calculus and 21% with shallow pockets [13]. Dental therapists and OHs are qualified to treat these periodontal conditions.

### 2.4. Oral Cancer

The prevalence of oral cancer has not declined over the past few years. Oral cancer has been associated with older men and, to a lesser degree, older females between the ages of 60 and 70 [17]. It must be noted that this prevalence does not include cancer of the tongue, gums and lips and is an underestimate of oral cancers in SA. Although tobacco use is still a common cause of oral cancer, there is a rising incidence of oropharyngeal cancer amongst men younger than 45 years old, which is associated with the Human Papilloma Virus (HPV). The risk of HPV related oropharyngeal cancer is nine-fold higher in individuals who are infected with HIV [18]. Whilst the prevalence of oral cancer is lower than other cancers, late diagnosis can significantly compromise the individual’s quality of life. It is essential that OHs and DTs are adequately trained to identify, diagnose and refer patients with suspected benign and malignant lesions.

## 3. Recommendations

We propose that most oral conditions could be addressed if the appropriate oral hygiene education and instructions were presented to parents of young children. Parents with the appropriate knowledge would presumably care for their children’s teeth and diet, possibly reducing the prevalence of dental caries. Educational campaigns could be implemented to deliver appropriate and relevant information to parents. Oral hygienists should be employed in ante-natal clinics to provide the necessary information. Nurses should also educate mothers on the importance of oral hygiene. We also propose that an oral health examination should be added to the “road to health” chart, which is a record of children’s routine immunizations. At these visits, nurses could offer dental screening and refer patients to oral health personnel if necessary.

Young children, especially between three and six years old, should receive regular dental screenings, oral hygiene instructions, and preventive services. These services should be offered before the age of six to help reduce the prevalence of caries, and subsequent sequelae in the primary dentition and early permanent dentition, as reported in previous studies [19].

Studies have reported that oral health could improve by utilizing community health workers to offer oral health information on the prevention of oral diseases [20]. This approach would ensure that oral health information is offered together with basic health care and allows for the improvement of communities’ awareness and health care practices.

### 3.1. Training of OHs

Given the current dental caries burden and personnel breakdown, dental schools should increase the number of training positions for OHs. The public sector should employ more OHs to help educate and inform mothers and children on the importance of good oral hygiene. Oral hygienists could also offer fissure sealants and brushing programmes in primary schools which could prevent the onset of dental caries.

Ideally, oral hygienists should be employed at primary health care (PHC) facilities to assist with screening and to receive children referred from PHC nurses, doctors and immunization clinics. Oral hygienists in PHC facilities could also provide services such as fissure sealants, preventive resin restorations, scaling, and polishing.

### 3.2. Training of Dental Therapists

There is a concern regarding the increased training of DTs given that there are limited posts in the public sector. Given the high burden of dental caries and subsequent treatment needs, it makes financial sense to train more DTs. The training program for DTs is shorter, three years, and the financial remuneration package for a DT is lower than that of a dentist. Despite the economic benefits, there are few posts for DTs in the public health sector. The government must create more posts to employ DTs, with a focus on prevention and education. Currently, most DTs are employed in the private sector because there are not enough posts, and DTs are poorly paid in the public sector. The government must address these concerns to attract and retain DTs in the public sector where they will be able to provide the bulk of dental care, namely dental extractions and restorations.

### 3.3. Future of Dentists

There is a debate in SA, that we should be training fewer dentists and more OHs and DTs. Given the disease burden, dentists seem over-qualified to deal with simple restorations and extractions. Most South Africans are unable to afford private dental care and use the public sector, which cannot cope with the huge demand for dental services. The private sector is often perceived to over-service patients and use unnecessary diagnostic tests and tools to diagnose dental conditions [21].

In SA, dentists have to complete a year-long compulsory community service program at accredited sites. Currently, anecdotal evidence suggests that dentists are being placed in facilities which do not have appropriate or sufficient equipment and materials. As a result, dentists are unable to offer any services at all or offer only limited dental services. We propose that it will be more cost effective to improve and increase the number of dental facilities and then place dentists appropriately. Community service dentists should be placed in areas where there is a relatively high need for dental services, and in facilities which have the basic equipment in a working condition. Optimal placement of dentists and DTs would improve access to dental services and oral health with the current supply of dentists that graduate every year.

## 4. Conclusions

An effective and efficient oral health care system requires a right number and mix of oral health personnel delivering appropriate quality care in places of greatest need to the people who need them most. South Africa has a high prevalence of oral diseases which could be prevented and treated by OHs and DTs in the public sector. The oral health personnel must be constantly reviewed and aligned with the current disease burden. Since the disease burden and demography is dynamic, the dental needs and requirements must constantly be determined and modified to best suit the South African needs. At present, the infrastructure at PHC facilities must be improved in order to accommodate and utilize community service dentists more efficiently. In the long run, SA requires more dental therapists and oral hygienists to be trained at dental institutions.

## Figures and Tables

**Figure 1 ijerph-16-01668-f001:**
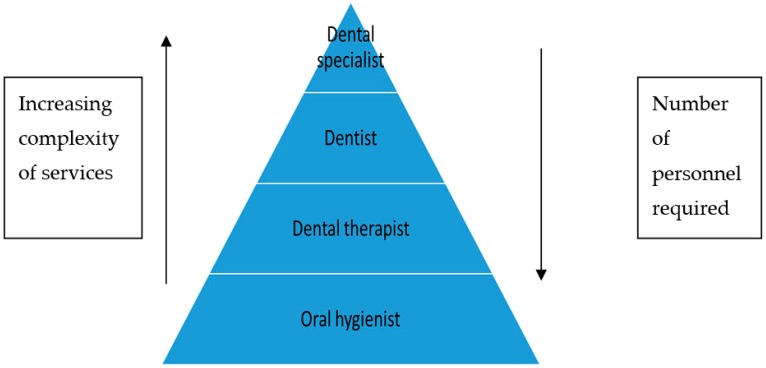
Breakdown of oral personnel in terms of delivery of services.

**Table 1 ijerph-16-01668-t001:** Number of dental specialists by area of specialization currently registered in SA (*N* = 481).

Type of Specialist	Number (%)
Maxillo-facial surgeons	144 (30)
Orthodontists	142 (30)
Prosthodontists	83 (17)
Periodontists	57 (12)
Community dentists	36 (7)
Oral pathologists	19 (4)
Total specialists	481 (100)

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
