# Peer review of "Human Resources for Oral Health Care in South Africa: A 2018 Update"

_ijerph, 2019, doi:10.3390/ijerph16101668_

Round 1

Reviewer 1 Report

This is an interesting paper but it needs a major review. The English needs work, there is much repetition in the paper, it needs to be formatted probably (see my suggestions below), and needs evidence to support some of the statements made.

Three times in the paper do you describe the typoes of dental  personel in South Africa. Once would be enough. This occurs because you have an introduction section, followed by a literature review. I suggest you combine the literature review and the introduction sections as one.

The methods section is very short and even then you mention the recommendations that you agian mention later in the paper. The only methods you have is tyo describe where you obtained the data used in the paper. This isn't really a scientific paper. It's more a discussion paper. If so, then a methods section is not needed.

The pie chart in figure doesn't add anything more than what is in the preceding paragraph. Hence, it is not needed.

The section from "3.3. Way forward" is really the discussion section of the paper. 

Author Response

ijerph-468064 authors’ response to reviewer 1

This is an interesting paper but it needs a major review. The English needs work, there is much repetition in the paper, it needs to be formatted probably (see my suggestions below), and needs evidence to support some of the statements made.

Three times in the paper do you describe the types of dental personnel in South Africa? Once would be enough. This occurs because you have an introduction section, followed by a literature review. I suggest you combine the literature review and the introduction sections as one.

We have combined the two sections as suggested.

The methods section is very short and even then you mention the recommendations that you agian mention later in the paper. The only methods you have is to describe where you obtained the data used in the paper. This isn't really a scientific paper. It's more a discussion paper. If so, then a methods section is not needed.

We have removed the Methods section

The pie chart in figure doesn't add anything more than what is in the preceding paragraph. Hence, it is not needed.

It has been removed

The section from "3.3. Way forward" is really the discussion section of the paper. 

We have changed it to Recommendations

Reviewer 2 Report

Strengths:

Thorough and clear descriptions of clinical oral health personnel.

The figures and tables are clear and easy to understand.

Figure 2 would be augmented by having a scale for the y axis, or clarify that this pyramid is for number of providers, complexity of services, etc.

Figure 3 would be improved with a correction of the scale on the x axis, and a description of the trendline.

Suggestions for revisions:

The paper is more of a description of current state and statement of aspiration, but it is not an evidence-based research paper.  It is not clear to the reviewers if this fits the IJERPH.

Since the author stated that most of the dental health care providers practice in the private sector, it would be beneficial to include evidence of relationship between increased staff and decreased disease burden.  Similarly, to detail the evidence of more OHs and DTs positively influencing dental outcomes . There are many aspirational statements in here that need more evidence to substantiate the claims, and make the paper more compelling.

Line 151-153—needs supporting evidence for  “probably indicated” and substantiation for the last sentence—is there any data to support that?

Lines 182-185 would be strengthened by citing literature about the decreased cost of care with earlier initiation of preventive care.

Lines 186-188 have 2 unconnected clauses; they should be separated.  The last sentence does not flow from the first 1 (should be 2); more needs to be inserted to connect and support the assertion at the end.

The paper has not supported enough of the claims to be able to make the conclusions in the conclusion.  Lines 232-234 are speculation.

Overall, this is a fine description of SA workforce that could be a great policy piece if it was supported more with evidence.

Minor Points:

Line 28—specify for all diseases and/or oral health specifically

 In Line 146, change “1989 till 2016” to “1989 to 2016.”

In Line 164, change “It must be noted however, that..” to “However, it must be noted that..”

In Lines 235-236, change sentence that starts as “SA also, in the long run, requires fewer” to “In the long run, SA requires fewer dentists.”

Line 162—refers to oral cancer “associated with older men”—what about women?  And “older” is refuted later in the paragraph.

In line 174 and later, there is reference to mothers.  Should this not be parent(s)? Especially if it is in  the “way forward” section.

South Africa is abbreviated SA in some places, but is not consistent throughout. (eg line 217)

Conclusion:

Line 211 states “There is a suggestion in SA, that we should be training fewer dentists and more OHs and DTs.”  If this idea is already suggested in SA, what is the author adding to the literature?

Author Response

Reviewer Two

Thorough and clear descriptions of clinical oral health personnel.

The figures and tables are clear and easy to understand.

Figure 2 would be augmented by having a scale for the y axis, or clarify that this pyramid is for number of providers, complexity of services, etc.

This has been done

Figure 3 would be improved with a correction of the scale on the x axis, and a description of the trendline.

This has been done

Suggestions for revisions:

The paper is more of a description of current state and statement of aspiration, but it is not an evidence-based research paper.  It is not clear to the reviewers if this fits the IJERPH.

Since the author stated that most of the dental health care providers practice in the private sector, it would be beneficial to include evidence of relationship between increased staff and decreased disease burden.  Similarly, to detail the evidence of more OHs and DTs positively influencing dental outcomes.

There are many aspirational statements in here that need more evidence to substantiate the claims, and make the paper more compelling.

Line 151-153—needs supporting evidence for  “probably indicated” and substantiation for the last sentence—is there any data to support that?

The reference has been included

Lines 182-185 would be strengthened by citing literature about the decreased cost of care with earlier initiation of preventive care.

This has been done

Lines 186-188 have 2 unconnected clauses; they should be separated.  The last sentence does not flow from the first 1 (should be 2); more needs to be inserted to connect and support the assertion at the end.

This has been reworded and changed accordingly.

The paper has not supported enough of the claims to be able to make the conclusions in the conclusion.  Lines 232-234 are speculation.

 This has been removed

Overall, this is a fine description of SA workforce that could be a great policy piece if it was supported more with evidence.

Minor Points:

Line 28—specify for all diseases and/or oral health specifically

This has been specified

 In Line 146, change “1989 till 2016” to “1989 to 2016.”

Done

In Line 164, change “It must be noted however, that..” to “However, it must be noted that..”

Done

In Lines 235-236, change sentence that starts as “SA also, in the long run, requires fewer” to “In the long run, SA requires fewer dentists.”

Done

Line 162—refers to oral cancer “associated with older men”—what about women?  And “older” is refuted later in the paragraph.

This has been reworded

In line 174 and later, there is reference to mothers.  Should this not be parent(s)? Especially if it is in  the “way forward” section.

It has been changed to parents

South Africa is abbreviated SA in some places, but is not consistent throughout. (eg line 217)

This has been changed

 Conclusion:

Line 211 states “There is a suggestion in SA, that we should be training fewer dentists and more OHs and DTs.”  If this idea is already suggested in SA, what is the author adding to the literature?

This suggestion is being discussed and as yet many academics and private practitioners are not fully supportive of the idea. Therefore, we have supported this idea and hope that it can be discussed more formally so that all stakeholders can be involved in the decision. 

Round 2

Reviewer 1 Report

This is a much better paper than the original version. With some relatively minor chnages iut should read well.

My largest concerns are:

1/ Under "3.1 Dental caries", you have new sentences, "The number of health personnel have increased yet the disease burden has continued to rise over the past few years. This could be  due to urbanization, an increase in the population of young children in the country and a lack of  clinical services in some areas." I'd like you to expand on how urbanization leads to an increased disease burden. I assume that increased urbanization leads to increased access to high sugar diets. 

2/ You use Figure 2 on the prevalence of dental caries to assert that there is an increasing prevalence of dental caries in South Africa. My concern is that this figure leaves you open to criticism by readers who may say it doesn't show an increasing trend. To show an increasing trend you need to have 95% confidence intervals in the chart so that the reader can see that the prevalnce 95% confidence intervals don't overlap between the years. Even then, the hypotehsised trend yelies only on three years (1989, 2002 & 2014). If I were in your shoes, I would simply remove Figure 2 as being unnecessary.

Minor Changes:

In the sentence "South Africa (SA), like many developing countries has a severe shortage of health personnel coupled with a high oral health disease burden" in the introduction, there should be a comma after vthe word "countries".

I hate the word "incentivized". May we change the sentence "Ideally, dental therapists should be encouraged and incentivized to working in the public sector to ensure that ...." in the introduction to "Ideally, dental therapists should be encouraged to work in the public sector to ensure that ...."?

Please change the sentence "If the bulk of dental disease, including dental caries, are prevented and treated at the primary health care level, the demand for specialized rehabilitative services should be very low" in the introduction to "If the bulk of dental disease, including dental caries, are prevented and treated at the primary health care level, the demand for specialized rehabilitative services should decrease". You are on dangerous ground predicting that the the demand would be very low in a country with increasing dental caries.

Author Response

This is a much better paper than the original version. With some relatively minor chnages iut should read well.

My largest concerns are:

1/ Under "3.1 Dental caries", you have new sentences, "The number of health personnel have increased yet the disease burden has continued to rise over the past few years. This could be due to urbanization, an increase in the population of young children in the country and a lack of clinical services in some areas." I'd like you to expand on how urbanization leads to an increased disease burden. I assume that increased urbanization leads to increased access to high sugar diets. 

This has been expanded on and I have included a reference to substantiate this claim.

2/ You use Figure 2 on the prevalence of dental caries to assert that there is an increasing prevalence of dental caries in South Africa. My concern is that this figure leaves you open to criticism by readers who may say it doesn't show an increasing trend. To show an increasing trend you need to have 95% confidence intervals in the chart so that the reader can see that the prevalence 95% confidence intervals don't overlap between the years. Even then, the hypothesised trend relies only on three years (1989, 2002 & 2014). If I were in your shoes, I would simply remove Figure 2 as being unnecessary.

We have removed the figure as suggested.

Minor Changes:

In the sentence "South Africa (SA), like many developing countries has a severe shortage of health personnel coupled with a high oral health disease burden" in the introduction, there should be a comma after the word "countries".

Done

I hate the word "incentivized". May we change the sentence "Ideally, dental therapists should be encouraged and incentivized to working in the public sector to ensure that ...." in the introduction to "Ideally, dental therapists should be encouraged to work in the public sector to ensure that ...."?

This has been reworded

Please change the sentence "If the bulk of dental disease, including dental caries, are prevented and treated at the primary health care level, the demand for specialized rehabilitative services should be very low" in the introduction to "If the bulk of dental disease, including dental caries, are prevented and treated at the primary health care level, the demand for specialized rehabilitative services should decrease". You are on dangerous ground predicting that the demand would be very low in a country with increasing dental caries.

Noted and changed.
